# Deficiency of mannose-binding lectin is a risk of *Pneumocystis jirovecii* pneumonia in a natural history cohort of people living with HIV/AIDS in Northern Thailand

**Kunio Yanagisawa** [1,2☯], **Nuanjun Wichukchinda** [3☯], **Naho Tsuchiya** [4], **Michio Yasunami** [5], **Archawin Rojanawiwat** [3], **Hidenori Tanaka** [6], **Hiroh Saji** [6], **Yoshiyuki Ogawa** [1], **Hiroshi Handa** [1], **Panita Pathipvanich** [7], **Koya Ariyoshi** [8]*, **Pathom Sawanpanyalert** [3¤]

1 Department of Hematology, Gunma University Hospital, Maebashi, Japan, 2 Infection Control and Prevention Center, Gunma University Hospital, Maebashi, Japan, 3 Department of Medical Sciences, Ministry of Public Health, Nonthaburi, Thailand, 4 Department of Preventive Medicine and Epidemiology, Tohoku Medical Megabank Organization, Tohoku University, Sendai, Japan, 5 Department of Laboratory Medicine, Saga-Ken Medical Centre Koseikan, Saga, Japan, 6 HLA Foundation Laboratory, Kyoto, Japan, 7 Day Care Centre, Lampang Hospital, Lampang, Thailand, 8 Department of Clinical Medicine, Institute of Tropical Medicine, Nagasaki University, Nagasaki, Japan

☯ These authors contributed equally to this work.
¤ Current address: Ministry of Science & Technology, Higher Education, Science, Research & Innovation
* kari@nagasaki-u.ac.jp

## Abstract

### Background

Mannose-binding lectin (MBL) plays a pivotal role in innate immunity; however, its impact on susceptibility to opportunistic infections (OIs) has not yet been examined in a natural history cohort of people living with HIV/AIDS.

### Methods

We used archived samples to analyze the association between MBL expression types and risk of major OIs including *Pneumocystis jirovecii* pneumonia (PCP), cryptococcosis, talaromycosis, toxoplasmosis, and tuberculosis in a prospective cohort in Northern Thailand conducted from 1 July 2000 to 15 October 2002 before the national antiretroviral treatment programme was launched.

### Results

Of 632 patients, PCP was diagnosed in 96 (15.2%) patients, including 45 patients with new episodes during the follow-up period (1006.5 person-years). The total history of PCP was significantly associated with low MBL expression type: high/intermediate (81/587, 13.8%), low (10/33, 30.3%) and deficient (5/12, 41.7%) (p = 0.001), whereas the history of other OIs showed no relation with any MBL expression type. Kaplan–Meier analysis (n = 569; log-rank p = 0.011) and Cox's proportional hazards model revealed that deficient genotype dramatically increased the risk of PCP, which is independent upon sex, age, CD4 count, HIV-1 viral

**Data Availability Statement:** All relevant data are within the manuscript and its Supporting Information files.

**Funding:** KY was financially supported by the grant of Joint Usage/Research Center on Tropical Diseases, Nagasaki University Institute of Tropical Medicine, Japan International Co-operation Agency (JICA) and Thai Ministry of Public Health for this work (2016-Ippan-4).http://www.tm.nagasaki-u.ac.jp/nekken/joint/files/h28reportbook.pdf.

**Competing interests:** The authors have declared that no competing interests exist.

load and hepatitis B and C status (adjusted hazard ratio 7.93, 95% confidence interval 2.19–28.67, p = 0.002).

## Conclusions

Deficiency of MBL expression is a strong risk factor determining the incidence of PCP but not other major OIs.

## Introduction

Among people living with human immunodeficiency virus (HIV) in Asia, *Pneumocystis jirovecii* pneumonia (PCP) is one of the most frequent opportunistic infections (OIs) in acquired immunodeficiency syndrome (AIDS)-defining diseases [1–4]. A cut-off level of <200 cells/μl for CD4 cell counts is well established as a risk factor for PCP [5]; therefore, CD4 cells probably play a pivotal role in the defense against *P. jirovecii* [6]. However, the incidence of PCP differs in various populations, such as a lower incidence in patients originating from sub-Saharan Africa compared with patients from Western origin [7], and CD4 cell counts in HIV patients are broadly distributed at the onset of PCP in the real world [8]. This variability raises the possibility that risk factors, e.g., host factors and/or causative pathogens, other than CD4 cell counts contribute to the development of PCP.

The *P. jirovecii* cell wall contains abundant glycoproteins including β-D glucan and mannose [6], and host pattern recognition receptors (PRRs) bind these glycoproteins via carbohydrates containing a C-type lectin-like domain, so-called C-type lectin receptors (CLRs) [9, 10]. They can be found as soluble forms and/or transmembrane receptors of various immune cells including macrophages, dendritic cells and neutrophils. RPRs/CLRs induce cytokine release and phagocytosis following interaction with fungus [9, 10]. Recently, the contributions of genetic variation of PRRs/CLRs to susceptibility to fungal infections have been reported [9, 10]. We speculate these genetic host factors play an important role in the development of PCP.

Mannose-binding lectin (MBL) is one of the well-studied soluble PRRs/CLRs, which plays a pivotal role in innate immunity against fungus [11]. The detailed protective mechanism is based on direct opsonization and activation of the complement system via the mannose-associated serine protease, which is involved in the lectin pathway [11–13]. Binding of MBL to *P. jirovecii* is followed by the activation of respiratory bursts and control of fungal spread [14]. We previously showed that supplementation of MBL enhances phagocytic reactions by macrophages in vitro [8]. However, the association between genetic variations in MBL and susceptibility to PCP has not yet been fully clarified.

Single nucleotide polymorphisms (SNPs) in the promoter region (rs11003125, H/L and rs7096206, X/Y), 5′-untranslated region (UTR) (rs7095891, P/Q) and exon 1 (rs1800450, B; rs1800451, C; and rs5030737, D) of *MBL2* gene are well known to influence MBL gene expression and structure [13]. We previously reported the association between *MBL2* genotypes and prevalence of PCP in HIV-infected patients in a cross-sectional observation of 53 patients in Japan [8]. However, information about polymorphisms and PRRs/CLRs affecting *Pneumocystis* recognition remain limited and their impact on susceptibility to AIDS-defining OIs has not been evaluated in a natural history cohort of patients with HIV/AIDS.

Previously, an observational cohort study was conducted at the HIV Clinic, Day Care Center (DCC) of Lampang Hospital in Northern Thailand, called the Lampang HIV cohort [1, 15–20]. The DCC was established to provide care and support for HIV-infected patients. Patient

recruitment started on July 6, 2000, and the follow-up was concluded at the date of induction of antiretroviral therapy (ART), patient death, or the last visit in October 2004.

Therefore, the aim of this study was to clarify the impact of MBL expression type defined by MBL2 genotypes on the risk of developing HIV/AIDS-related OIs, especially PCP, in a natural history HIV cohort in northern Thailand.

## Material and methods

### Design and population

In total, 755 ART naïve patients recruited in the previously descrived Lampang HIV natural history cohort were re-analyzed [21]. Briefly the recruitment of this cohort was done from 1 July 2000 to 15 October 2002 before the national antiretroviral treatment programme was launched. All adult (aged > 18 years) HIV-infected individuals attending the HIV clinic who were ART-naïve at the first visit were approached. Clinical data of individuals, who were followed up for at least two time points, were used for the current longitudinal analysis in our group [22]. Baseline clinical data were collected when participants were registered in this cohort. The history of OIs was noted within the follow-up period. The diagnosis of OIs was based on laboratory data and typical findings from radiological images, following the Ministry of Public Health, Thailand. *National guidelines for the clinical management of HIV infection in children and adults.* (Sixth edition); 2000 [23].

The prophylactic and therapeutic interventions agains OIs were based on the same guidelines [23]. Briefly, for primary prophylaxis, patients with a CD4 count <200/µl were given two double-strength tablets of trimethoprim/sulfamethoxazole (TMP/SMX; 80 mg TMP and 400 mg SMX) orally once daily for prophylaxis against PCP. The same regimen was administered to prevent toxoplasmosis when the CD4 count was <100 /µl. Fluconazole 200 mg orally once daily or 400 mg once a week was given for prophylaxis against cryptococcosis when the CD4 cell count was <100 /µlL. No primary prophylaxis for TB or Mycobacterium avium complex (MAC) infection was given in this study.

### Experimental procedures

*MBL2* **genotyping.** *MBL2* genotyping was performed by using a multiplexed microsphere suspension array-based platform, Luminex xMAP™, as previously described [24]. Briefly, genomic DNA was extracted from peripheral blood using a standard phenol-chloroform method, and the *MBL2* promoter region and 5′-UTR (-619, -290, and -66), a 677-bp fragment was amplified with 5′ biotinylated primers and then directly hybridized with six oligonucleotide probes specific for each allele of the corresponding biallelic SNP, which was immobilized on the microsphere beads. *MBL2* exon 1 haplotypes were determined by PCR-preferential homo-duplex formation assay, in which unlabeled PCR products were hybridized with eight double-labeled (biotin and specific capture sequence) 65-bp double-stranded standards, corresponding to each of the theoretically possible exon 1 cdn52-cdn54-cdn57 haplotypes: CGG, CAG, TGG, CGA, CAA, TAG, TAA, and TGA. These SNPs comprise seven established haplotypes: HYPA, LYQA, LYPA, LXPA, HYPD, LYPB, and LYQC.

There are no universal definitions of category based on *MBL2* genotype associated with plasma MBL concentrations and functional complement activity. In this study, we defined four MBL expression types: deficient, low, intermediate and high, which based on the classification reported by Chalmers et al. [25] (**Table 1**). We modified their classification because "low" defined by them includes both complete deficiency and relatively low expression; therefore, we distinguished and defined them as "deficient" and "low".

**Table 1. Definition of MBL expression type.**

| Haplotype | Genotype | Combination | MBL expression type[a] |
|---|---|---|---|
| HYPD/LYQC/LYPB | O | YO/YO | deficient |
| LXPA | XA | XA/YO | low |
| LYPA/LYQA | YA | YA/YO | intermediate |
| HYPA | YA | XA/XA | high |
| | | YA/XA | high |
| | | YA/YA | high |

[a]MBL expression type was defined by abbreviated genotypes and combinations based on each haplotype.

**Measurements of plasma MBL concentrations.** To confirm the validation of MBL expressions described above, we tested 273 stored samples; selection process was restricted by the availability of stored plasma and samples were more preferentially selected from patients with a history of PCP and patients with low MBL expression type rather than high MBL expression type. These were linked concentrations to *MBL2* genotype of each patient. Plasma MBL concentrations were determined using the MBL Oligomer ELISA kit (BioPorto Diagnostics, Hellerup, Denmark) according to the manufacturer's instructions.

## Statistical analysis

The Mann–Whitney U test was used to compare the medians of continuous variables. The chi-square test was used to compare categorical variables and to assess the significance of deviations from Hardy–Weinberg equilibrium (HWE). Kaplan–Meier analysis and log-rank test were performed to estimate the risk of PCP during the follow-up period. Cox's proportional hazards models were performed to analyze the risk of PCP as the dependent variable. Adjusted hazard ratios (aHR) and 95% confidence intervals (CIs) were determined, and $p<0.05$ was considered significant. SPSS version 25 (IBM, Chicago, IL, USA) and STATA version 14.0 (StataCorp, College Station, TX, USA) were used to perform analyses.

## Ethical statements

This study was approved by the Gunma University Ethical Review Board for Medical Research Involving Human Subjects (#150018, March 16, 2016). Furthermore, the pilot study registered at the Thai National Institute of Health received approval from the Ethics Committee of the Institute for Development of Human Research Protections for "studies of host genetic, immunological, virological and co-infection factors associating with HIV/AIDS". This study is the secondary use of the unlinked anonymous sample collected in Lampang HIV cohort, and the authors does not obtain the information that can identify an individual before the analysis of data.

## Results

In total, 632 individuals with the median (IQR) follow-up period of 471.5 (187.5–913) days were recorded during the 1006.5 person-years of observation. We successfully determined *MBL2* genotypes of 632 individuals and found that the majority (92.9%) of individuals had high/intermediate (69.6%/23.3%) MBL expression, and the others had low (5.2%) and deficient (1.9%) MBL expression. All studied SNPs were in HWE. Ninety-six (15.2%) patients had a past and/or present history of PCP: 60 (9.5%) patients at the time of recruitment and an additional 45 (7.1%) patients during the follow-up period. Nine patients developed PCP twice

**Table 2. MBL expression type and characteristics of all study patients.**

|  | Total | deficient | low | high+int | *p* |
|---|---|---|---|---|---|
|  | (n = 632) | (n = 12) | (n = 33) | (n = 587) |  |
|  | 100.0% | 1.9% | 5.2% | 92.9% | 0.409 |
| Male sex | 265 (41.9%) | 7(58.3%) | 1 (33.3%) | 247(42,1%) | 0.311 |
| Age (range) | 33 (15–63) | 33(28–47) | 33 (18–60) | 33(15–63) | 0.732 |
| CD4+ cell count/μl | 154 | 241 | 90 | 156 | 0.518 |
| (range) | (0–1191) | (10–412) | (0–762) | (0–1191) |  |
| $\log_{10}$VL (copies/ml) | 5.19 | 4.63 | 5.23 | 5.19 | 0.858 |
| (range) | (2.60–6.72) | (2.60–6.28) | (2.60–6.07) | (2.60–6.72) |  |
| PCP | 96 (15.2%) | 5(41.7%) | 10 (30.3%) | 81(13.8%) | 0.001† |
| Cryptococcosis | 83 (13.2%) | 1(8.3%) | 6 (18.2%) | 76(13.0%) | 0.609 |
| Talaromycosis | 49 (7.8%) | 0(0.0%) | 2 (4.1%) | 47(8.0%) | 0.550 |
| Toxoplasmosis | 30 (4.8%) | 1(8.3%) | 3 (9.1%) | 26(4.5%) | 0.4.4 |
| Tuberculosis | 118 (18.7%) | 3(25.0%) | 5 (15.2%) | 110(18.8%) | 0.745 |
| HBsAg | 70(11.2%) | 3(25.0%) | 4(12.1%) | 63(10.9%) | 0.304 |
| HCVAb | 24(3.8%) | 1(8.3%) | 0(0.0%) | 23(4.0%) | 0.368 |

Abbreviations: high+int, high and intermediate; VL, HIV-RNA viral load; PCP, *Pneumocystis jirovecii* pneumonia. Kruskal-Wallis test was used to compare the medians of continuous variables, and the chi-square test was used to compare categorical variables.

†statistically significant (*p*<0.05).

before and after the recruitment. The total history of PCP (n = 96) was significantly associated with MBL expression type (*p* = 0.001): 81 of 587 (13.9%) as high/intermediate vs 10 of 33 (30.3%) as low and 5 of 12 (41.7%) as deficient (**Table 2**). However, the total histories (before and/or after the recruitment) of other major OIs including cryptococcosis, talaromycosis, toxoplasmosis, and tuberculosis showed no relation with any MBL expression type (**Table 2**). There were no differences in the history of PCP between high and intermediate MBL expression types.

New incidences of PCP episodes were plotted according to MBL expression type by Kaplan–Meier analysis (**Fig 1**). The deficient expression group was significantly associated with the incidence of PCP compared to low and high/intermediate group (n = 569, log-rank, *p* = 0.011, **Fig 1A**). Furthermore, the association was remained either in subgroup of CD4<200/μl or ≧200/μl (n = 293 and 274, log-rank, *p* = 0.030 and 0.021, respectively, **Fig 1B and 1C**) although patients with baseline CD4<200/μl had been taken TMP/SMX, it can be a strong confounder of PCP onset.

Univariate and multivariate Cox's proportional hazards models were fitted to identify predictors of PCP. Covariates included sex, age, CD4 count, viral load ($\log_{10}$ VL), and MBL expression type. CD4 count was also categorized into ≧200/μl or <200/μl in consideration of PCP prophylaxis by TMP/SMX. PCP cases were significantly associated with deficient type of MBL expression and CD4 count <200/μl, independent of sex, age, and $\log_{10}$VL (aHR 7.93, 95% CI 2.19–28.67, *p* = 0.002; aHR 3.57, 95% CI 1.51–8.45, *p* = 0.004, respectively) (**Table 3**).

Distributions of MBL concentrations measured in plasma samples (n = 273) are shown in **Fig 2**. Median plasma concentrations were well associated with the each MBL expression type: 0.0 ng/ml in deficient, 25.3 ng/ml in low, 612.2 ng/ml in intermediate, and 2954.7 ng/ml in high with significant differences between two pairs or whole four types in each other (**Fig 2**, *p*<0.001). Therefore, the validity of *MBL2* genotypes was confirmed with the phenotypes in this study.

a. All (n=569), log-rank *p*=0.011

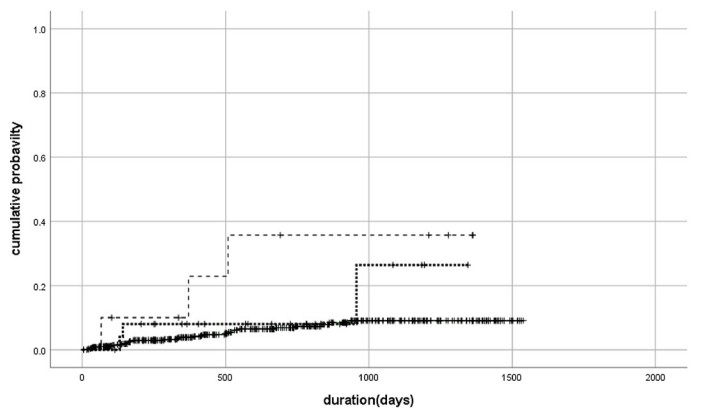

c. CD4≧200 (n=274), log-rank *p*=0.021

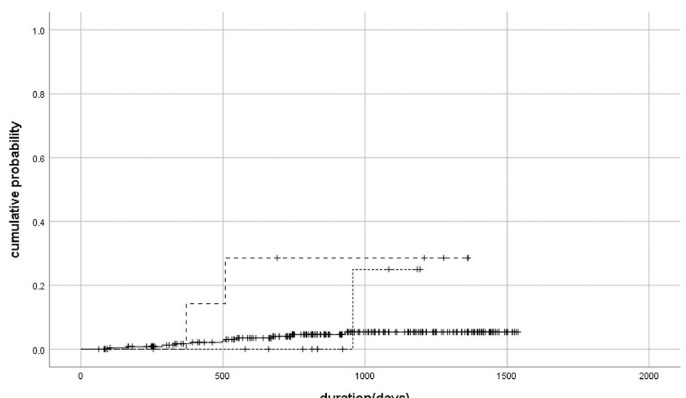

b. CD4<200 (n=293), log-rank *p*=0.030

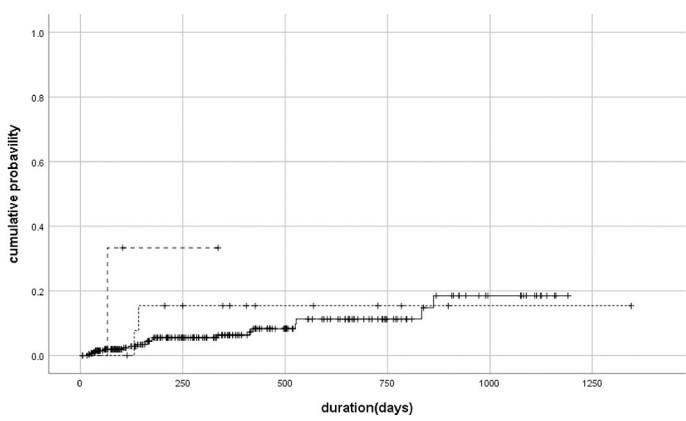

**Fig 1. Cumulative probability curves of PCP during follow-up periods.** Kaplan–Meier curves for registered *Pneumocystis jirovecii* Pneumonia (PCP) patients. (a) Total patients (n = 569), (b) baseline CD4 count <200/μl (n = 293), and (c) baseline CD4 count ≧200/μl (n = 274). Dashed line indicates deficient expression type of Mannose-binding Lectin (MBL), dotted line indicates low and solid line indicates high/intermediate MBL expression types combined. The log-rank test was used to compare group.

**Table 3. Univariate and multivariate Cox's proportional hazards models.**

| n = 567 | | Crude HR | 95% CI | *p* | aHR | 95% CI | *p* |
|---|---|---|---|---|---|---|---|
| Male sex | | 1.26 | 0.63–2.50 | 0.512 | 0.82 | 0.38–1.76 | 0.604 |
| Age | | 0.96 | 0.91–1.01 | 0.143 | 0.94 | 0.89–1.00 | 0.058 |
| CD4 count (/μl) | <200 | 3.31 | 1.64–6.66 | 0.001† | 3.57 | 1.51–8.45 | 0.004† |
| log₁₀VL (copies/ml) | | 1.70 | 1.09–2.65 | 0.020† | 1.26 | 0.75–2.10 | 0.379 |
| MBL expression type | deficient | 4.90 | 1.50–16.08 | 0.009† | 7.93 | 2.19–28.67 | 0.002† |
| | low | 1.86 | 0.57–6.10 | 0.305 | 1.52 | 0.45–5.15 | 0.502 |
| HBsAg | | 1.89 | 0.73–4.90 | 0.192 | 1.48 | 0.53–4.08 | 0.452 |
| HCVAb | | 1.41 | 0.34–5.88 | 0.640 | 1.60 | 0.36–7.11 | 0.534 |

Abbreviations: HR, hazard ratio; aHR, adjusted hazard ratio; CI, confidence interval; VL, HIV RNA viral load

[*1] referred to CD4≧200

[*2] referred to intermediate and high combined.

†statistically significant (*p*<0.05).

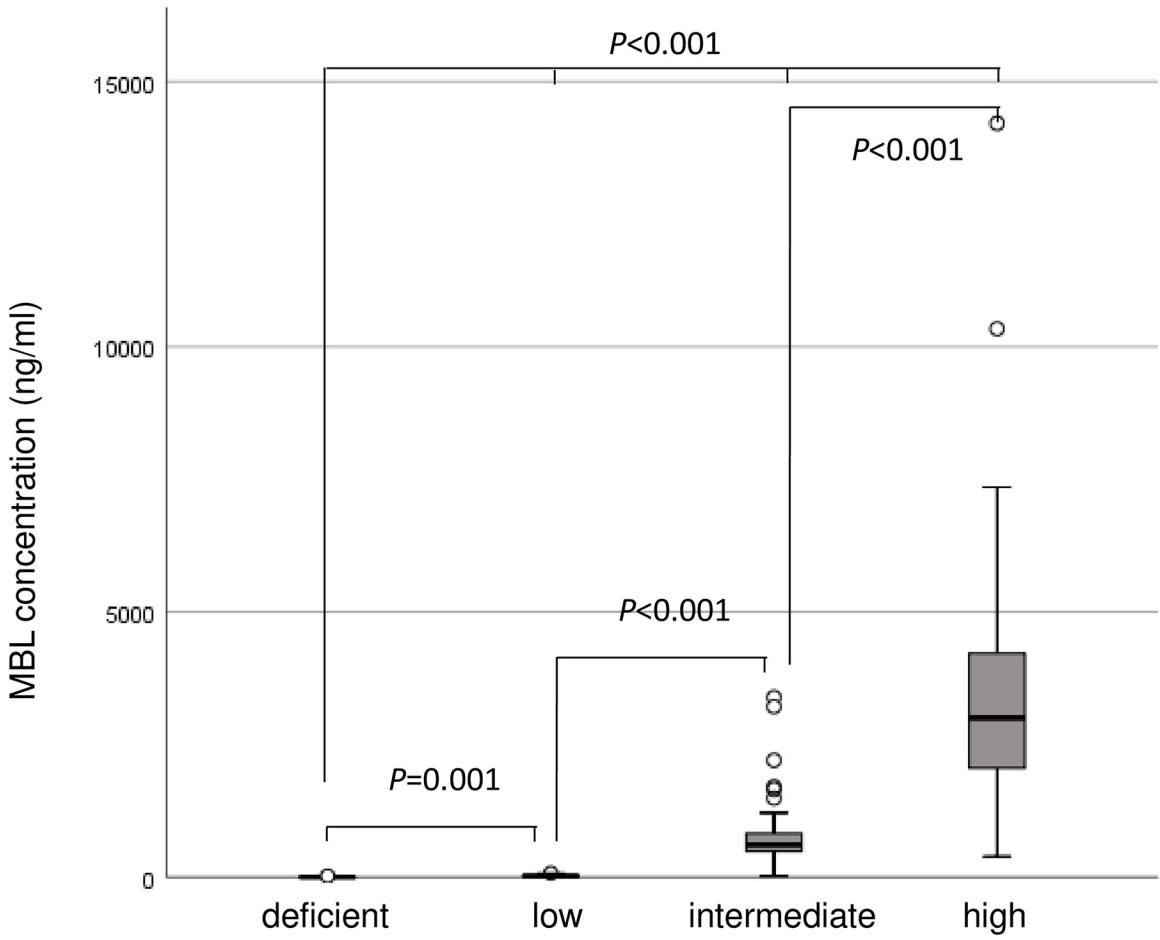

**Fig 2. Box plots of plasma MBL concentrations of patients (*n* = 273) according to MBL expression type.** Box plots of plasma Mannose-binding Lectin (MBL) concentrations of patients (n = 273) according to MBL expression type. The Mann–Whitney U test and Kruskal–Wallis test were used for analyzing the differences of plasma MBL concentrations in each MBL expression type.

## Discussion

Little has been known about host genetic polymorphisms affecting its sensitivity against PCP other than our previous study in limited number of Japanese populations [8, 10]. To our knowledge, this presenting data is the first cohort study demonstrating that MBL expression type defined by *MBL2* genotypes are significantly associated with the incidence of PCP. A recent study performed in a large cohort in Switzerland did not observe an association between MBL expression and PCP in patients whose CD4 cell counts were <200 cells/μl [26]. The authors also showed that CD4 cell count is one of the strongest risk factors of PCP. In contrast, our study showed that deficient MBL expression can be a strong risk of PCP in all followed-up patients as well as the suppression of CD4 cells<200/μl. Although the analyzed patients with CD4 <200/μl were on PCP prophylaxis, their clinical course was not influenced by ART, because the presenting study applied archive samples and data collected before ART was widely introduced in northern Thailand [21]. For this reason, we believe that our finding mirrors the real impact of MBL genetic variation on the host susceptibility to PCP, which may answer our primary question of why some people with advanced HIV infection develop PCP while others do not.

Our results indicate only PCP incidence was associated with MBL expression type, whereas other representative OIs such as cryptococcosis, talaromycosis, toxoplasmosis, and tuberculosis were not. This may contradict a previous study that showed an association between *MBL2* genotype and incidence of fungal infections, including *Cryptococcus*, *Aspergillus*, and *Candida* infections [11]. It has been reported that a number of pathogen-associated molecular patterns (PAMPs) on the surface of fungi, such as β-D glucan, O/N-linked mannan, chitin, DNA and RNA, have interactions with PRRs of host cells [9]. MBL binds glycosylated ligands of PAMPs with its carbohydrate recognition domain [27]. Our study demonstrated the relation of MBL and *Pneumocystis* among numbers of combinations between host PRRs and fungal PAMPs based on the analysis of clinical outcomes.

Definitions of *MBL2* genotype and/or MBL expression type have not yet been standardized. The "low" classification defined by Chalmers et al. [25] includes both a complete deficiency of MBL and relatively low expression. If we analyzed our data based on their classification, the association between incidence of PCP and "low" type was disappeared in subgroup of CD4<50/μl (**S1 Fig**). Furthermore, the classification defined by Ou et al. (including "deficient/low/high", which is equivalent to "low/intermediate/high" described above), seems that homogenous combination of mutant haplotype (YO/YO, refer to **Table 1**) and heterogenous (XA/YO) were confused into same classification to "deficient" [28]. Therefore, we remade the classification by distinguishing deficient and low expression which means homogenous haplotype (YO/YO) or heterogenous (XA/YO). The validity of these modified classification was verified by the measurement of plasma MBL concentration with gradual increasing of them according to the expression types. However, considerable overlap in plasma MBL concentration was observed in the presence or absence of PCP (**S2 Fig**), although MBL concentrations were clearly differentiated by expression type. These findings suggest there are other unknown factors influencing the incidence of PCP, which overcome the influence of MBL when patients are highly immunosuppressed.

Also, we have to consider racial differences can influence host defense mechanisms based on genotype heterogeneity [29, 30]. It has been reported that a number of major SNPs are associated with susceptibility of fungal infections and diseases, including *MBL2*, *TLR1/4/6/9*, *CARD9*, *CXCL2*, *DECTIN1*, *IL4/10/15/2* [31], and HLA [30]. Therefore, molecular interactions between pathogens and hosts should be discussed in consideration of the geographical area where the study was performed. We confirmed the associations between MBL and PCP both in Japanese [8] and Thai populations; thus, this biological mechanism appears to be true in Asian populations. Further study is warranted to demonstrate this association in Africa, where the incidence of PCP is lower [7].

Several limitations were included in this study. Firstly, we did not investigate genetic variations of other PRRs associated with fungal infections in this study. Furthermore, the possibilities of differences in *Pneumocystis* species or in pattern of surface molecules in Japan and Thailand populations were not investigated. However, the genetic diversity of *Pneumocystis* was limited in a multicenter study conducted in Europe [32]. Although MBL protein synthesis and release can be affected by liver function, levels of interleukin-6, growth and thyroid hormones [33], these data were not available. However, we have successfully shown the risk of deficient and low MBL expression type by Cox's proportional hazards models in consideration with seropositivity of HBV and HCV. It may partially reflect the confounders associated with liver function. Finally, we could not evaluate the association between MBL phenotype and the onset of other OIs. Also, we could not confirm the changes in plasma MBL concentrations between different collection days, because all of plasma samples were not available.

In summary, we firstly showed that deficiency of MBL expression is a strong risk factor determining the incidence of PCP, but not other major OIs, in HIV natural history cohort.

## Supporting information

**S1 Fig. Cumulative probability curves of PCP during follow-up periods (CD4<50/μl) his Kaplan-Meire curve shows the cumulative provability of PCP, if they are categorized by three MBL expression type (low/intermediate/high) and selected by their CD4 cells <50/μl (n = 171).** The solid line means "low", and dashed line means intermediate and high combined. The difference between "low "vs "int + high" were analyzed by log-rank test; however, there were no significance ($p = 0.44$).
(DOCX)

**S2 Fig. Box plots of plasma MBL concentrations with or without PCP during the follow-up period (n = 231).** MBL concentrations were plotted in two group of patients if they have a new episode of PCP (n = 19, plasma MBL was 0.0–4683.4/ng/ml, median 2008.9 ng/ml) or not (n = 212, 0.0–14213.3, median 2057.8 ng/ml) during the follow-up period. The median of each two group was tested by Mann-Whilney U test; however, there were no significance ($p = 0.679$).
(DOCX)

**S1 Dataset.**
(XLS)

## Acknowledgments

We thank Dr. Kunihiko Hayashi for providing kind assistance with statistical analysis. We also thank Ms. Maneeratt for providing support during the visit and research meeting in Thailand.

## Author Contributions

**Conceptualization:** Kunio Yanagisawa, Naho Tsuchiya, Michio Yasunami, Archawin Rojana-wiwat, Hidenori Tanaka, Hiroh Saji, Yoshiyuki Ogawa, Hiroshi Handa, Panita Pathipvanich, Koya Ariyoshi, Pathom Sawanpanyalert.

**Data curation:** Kunio Yanagisawa, Naho Tsuchiya, Panita Pathipvanich.

**Formal analysis:** Kunio Yanagisawa, Nuanjun Wichukchinda, Naho Tsuchiya, Koya Ariyoshi.

**Funding acquisition:** Kunio Yanagisawa.

**Investigation:** Kunio Yanagisawa, Nuanjun Wichukchinda, Naho Tsuchiya, Archawin Roja-nawiwat, Panita Pathipvanich, Koya Ariyoshi.

**Methodology:** Kunio Yanagisawa, Naho Tsuchiya.

**Project administration:** Kunio Yanagisawa, Hidenori Tanaka, Hiroh Saji, Panita Pathipvanich.

**Software:** Naho Tsuchiya.

**Supervision:** Naho Tsuchiya, Yoshiyuki Ogawa, Hiroshi Handa, Koya Ariyoshi, Pathom Sawanpanyalert.

**Validation:** Nuanjun Wichukchinda, Naho Tsuchiya, Michio Yasunami, Archawin Rojanawi-wat, Hidenori Tanaka, Hiroh Saji, Panita Pathipvanich, Koya Ariyoshi, Pathom Sawanpanyalert.

**Writing – original draft:** Kunio Yanagisawa.

**Writing – review & editing:** Nuanjun Wichukchinda, Naho Tsuchiya, Koya Ariyoshi.

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
