## [Decision Letter · Decision Letter 0]

11 Aug 2020

PONE-D-20-20525

Association between Mannose-binding Lectin Expression and Risk of Pneumocystis jirovecii Pneumonia in People Living with HIV/AIDS in Northern Thailand

PLOS ONE

Dear Dr. Yanagisawa,

Thank you for submitting your manuscript to PLOS ONE. After careful consideration, we feel that it has merit but does not fully meet PLOS ONE’s publication criteria as it currently stands. Therefore, we invite you to submit a revised version of the manuscript that addresses the points raised during the review process. Please pay close attention to the concerns of reviewer # 1 with regards to statistical analysis and the role of other confounders. There is also a minor issue raised by reviewer # 2 that you and your co-authors should be able to readily address. This is a nice and well performed study.

We look forward to receiving your revised manuscript.

Kind regards,

Aftab A. Ansari, PhD

Academic Editor

PLOS ONE

Journal Requirements:

2. Please provide the following details about the stored biological samples used in your study:

(1) source of the samples

(2) whether the samples were completely de-identified before researchers accessed the samples.

We note that you state that "stored samples derived from the Lampang HIV cohort in “unlinked anonymous” pattern can be used in subsequent analyses without requiring additional ethical approval." Please clarify whether the samples were "unlinked anonymous" before you accessed them. Please add this information to the methods section and your ethics statement in your online submission form.

Reviewers' comments:

Reviewer's Responses to Questions

**Comments to the Author**

1. Is the manuscript technically sound, and do the data support the conclusions?

Reviewer #1: Partly

Reviewer #2: Yes

2. Has the statistical analysis been performed appropriately and rigorously? 

Reviewer #1: I Don't Know

Reviewer #2: Yes

3. Have the authors made all data underlying the findings in their manuscript fully available?

Reviewer #1: Yes

Reviewer #2: Yes

4. Is the manuscript presented in an intelligible fashion and written in standard English?

Reviewer #1: Yes

Reviewer #2: Yes

5. Review Comments to the Author

Reviewer #1: The manuscript by Yanagisawa and colleagues describes the association between mannose-binding lectin and susceptibility to Pneumocystis jiroveccii pneumonia (PCP) in HIV+ ART-naive patients in Northern Thailand. PCP was diagnosed in 15% of patients and the authors report MBL expression to be associated with PCP onset in patients with CD4 counts > 50 /ul.

The study is straightforward, and the data are well presented. However, the Kaplan Meir curves showing decreased likelihood of PCP over time is confusing. Also, the lower risk of PCP incidence in patients with CD4 counts < 50/ul is unclear. Were other important confounders not accounted for such as susceptibility to other OIs? Another caveat is that presence of coinfections impacting liver function could impact MBL levels and this confounder was not controlled for.

Reviewer #2: The manuscript submitted by Yanagisawa and colleagues addresses the role of mannose binding lectin to susceptibility to Pneumocystis jirovecii infection. Specifically, they address the role of genotypes associated with MBL deficiency in the development of PCP in HIV infected subjects. The link between MBL deficiency and infection is an important subject and of significant interest to a broad array of biomedical and public health researchers. Susceptibility to PCP among HIV infected subjects is well-known and provides an opportunity to investigate this important question.

The authors utilize a special cohort of HIV infected subjects from Northern Thailand for which samples were taken in a time period prior to the administration of ART. Such samples are rare and valuable.

The authors carry out an MBL genotypic analysis of samples from the cohort, that allows them to identify subjects with high, intermediate and low MBL genotypes. This analysis is validated by measuring MBL levels in a subset of subjects by direct measurement of MBL. The association is strong. This analysis allows for identification of any correlations between PCP and MBL genotype.

Such an analysis is however nontrivial. HIV infected subjects encompass a range of immunocompetence, which itself can have a profound impact on opportunistic infection. The authors correctly address this variable by evaluating the relationship between PCP, MBL genotype and blood CD4 counts. CD4 counts are a strong measure of immune dysfunction in HIV infected subject and is significantly associated with PCP. Indeed, this variable has the potential to mask any effect of MBL genotype on susceptibility. In fact, for subjects with a CD4 count <50 the association between MBL genotype and PCP infection was lost. However, for subjects with CD4 counts <200 they identified strong and convincing correlation between PCP and genotypes associated with MBL deficiency.

I found this manuscript to be clearly presented. It was well organized and a pleasure to read. The subject is an interesting one. Numerous questions remain in relation to MBL deficiency and infection. I expect that these researchers will continue to contribute to this subject area.

The only suggestion that I might offer involves the distinction between MBL low genotypes and MBL deficiency. Although the authors demonstrate a very strong correlation between genotype and deficiency, their analysis of susceptibility to PCP is with the genotype, not the phenotype. Perhaps they could make this distinction in their concluding remarks.

6. PLOS authors have the option to publish the peer review history of their article (what does this mean?). If published, this will include your full peer review and any attached files.

Reviewer #1: No

Reviewer #2: No

---

## [Author Response · Author response to Decision Letter 0]

30 Oct 2020

To reviewer #1: 

“The manuscript by Yanagisawa and colleagues describes the association between mannose-binding lectin and susceptibility to Pneumocystis jiroveccii pneumonia (PCP) in HIV+ ART-naive patients in Northern Thailand. PCP was diagnosed in 15% of patients and the authors report MBL expression to be associated with PCP onset in patients with CD4 counts > 50 /ul. The study is straightforward, and the data are well presented. However, the Kaplan Meir curves showing decreased likelihood of PCP over time is confusing. “

Thank you for your suggestion. The Y-axis of Kaplan-Meier curves in Fig1 in the first manuscript may give misunderstanding that the incidence of the PCP looks to be decreased over time as you mentioned. In the revised manuscript, we modified Fig1. that Y-axis show the increase of PCP incidence.

“Also, the lower risk of PCP incidence in patients with CD4 counts < 50/ul is unclear. ”

Thank you for your suggestion. In this study, all patients having baseline CD4<200/µl were taken TMP/SMX for the PCP prophylaxis, so that the incidence of PCP in patients with CD4<50/µl group was modified by this medication. However, after the re-categorization of MBL expression type in the revised manuscript (deficient/low/intermediate/high), the cut-off level of CD4 counts below or over 50/µl were not a critical factor of PCP. Therefore, we added Fig 1. b (CD4<200) and c (≧200) in revised version to clarify the impact of PCP prophylaxis that they have been taken or not. As a result, we confirmed that deficient MBL expression type was an independent risk factor of PCP incidence whether CD4 counts were below nor over 200/µl through K-M analysis, univariate and multivariate Cox’s proportional hazard analysis (P10, line 204-210; P11, line 220-226). 

“Were other important confounders not accounted for such as susceptibility to other OIs? Another caveat is that presence of coinfections impacting liver function could impact MBL levels and this confounder was not controlled for.”

Thanks for your suggestion. Because the data associated with liver function (i.e. liver transaminase, prothrombin time and choline-esterase activity) and all of plasma samples were not available, we could not conduct the analysis including these confounders. We also agree that MBL synthesis can be affected by many other confounders including liver functions, co-infections such as hepatitis B and C virus and other OIs. In consideration of these concerns, we retried cross-sectional, univariate and multivariate analysis including HBsAg and HCV-Ab as additional confounding variables. As a result, deficient expression type remained as a significant risk factor of PCP histories (P9, line 190-193, Table 2), and its incidence (P11, line 220-226, Table 3). We believe these data show the critical role of MBL expression type at the onset of PCP upon the co-infection of hepatitis B and C virus.

To reviewer #2: 

The only suggestion that I might offer involves the distinction between MBL low genotypes and MBL deficiency. 

As you mentioned, patients classified “low” in our first manuscript includes complete deficiency to relatively low expression. Therefore, their plasma MBL concentrations distributed between undetectable to relatively low levels (shown in Fig 2). We re-analyzed our data according to your kind advice that we should distinguish the deficiency and low. As a result, we achieved more impressive results. Briefly, we confirmed that MBL deficiency associated with PCP histories (Table 2), cumulative provability (Fig 1) and incidence independently upon sex, age, CD4 and log10VL (Table 3). We appreciate your great advice and believe that our revised data improved to show the association between MBL expression type and PCP onset more clearly through the new manuscript.

Although the authors demonstrate a very strong correlation between genotype and deficiency, their analysis of susceptibility to PCP is with the genotype, not the phenotype. Perhaps they could make this distinction in their concluding remarks. 

As you mentioned, we agree the limitation that we could not analyze the MBL phenotype and the PCP onset, because all of plasma samples were not available. Therefore, we added this discussion that further studies are needed to clarify the association of MBL phenotype and PCP onset (P15, line 314-318).

Response to Journal Requirements:

We apologize that first manuscript was not suitable for the PLOS ONE format. We have made the revised manuscript according to the instruction included in the URL you provided, then please confirm it again.

2. Please provide the following details about the stored biological samples used in your study:

(1) source of the samples　(2) whether the samples were completely de-identified before researchers accessed the samples. We note that you state that "stored samples derived from the Lampang HIV cohort in “unlinked anonymous” pattern can be used in subsequent analyses without requiring additional ethical approval." Please clarify whether the samples were "unlinked anonymous" before you accessed them. Please add this information to the methods section and your ethics statement in your online submission form.

Thanks for your suggestion. (1) We used the genomic DNA extracted by the standard protocol of phenol-chloroform method from the peripheral blood of registered patients. We clarified this preparation steps in the revised manuscript (P6, line 128-129). (2) This study was performed by the second use of anonymity samples and information provided in Lampang HIV cohort. Before the analysis and manuscript preparation, authors have not been provided any information that can identify the individuals registered in cohort. We added this point in ethical statement (P8, line 177-179), and input in the submission form.

3. PLOS requires an ORCID iD for the corresponding author in Editorial Manager on papers submitted after December 6th, 2016. Please ensure that you have an ORCID iD and that it is validated in Editorial Manager. To do this, go to ‘Update my Information’ (in the upper left-hand corner of the main menu), and click on the Fetch/Validate link next to the ORCID field. This will take you to the ORCID site and allow you to create a new iD or authenticate a pre-existing iD in Editorial Manager. Please see the following video for instructions on linking an ORCID iD to your Editorial Manager account: 

https://www.youtube.com/watch?v=_xcclfuvtxQ

I (Kunio Yanagisawa) have already applied my information to ORCID form that you indicated.

We apologize our lack of understanding that the description of "data not shown " is not accepted. We deleted this description, then the data necessary to make discussion was included in supportive information at the end of revised manuscript.

---

## [Editor Report · Decision Letter 1]

3 Nov 2020

Deficiency of Mannose-binding Lectin is a Risk of Pneumocystis jirovecii Pneumonia in a Natural History Cohort of People Living with HIV/AIDS in Northern Thailand

PONE-D-20-20525R1

Dear Dr. Yanagisawa,

We’re pleased to inform you that your manuscript has been judged scientifically suitable for publication and will be formally accepted for publication once it meets all outstanding technical requirements.

Kind regards,

Aftab A. Ansari, PhD

Academic Editor

PLOS ONE
---

## [Editor Report · Acceptance letter]

24 Nov 2020

PONE-D-20-20525R1 

Deficiency of Mannose-binding Lectin is a Risk of *Pneumocystis jirovecii* Pneumonia in a Natural History Cohort of People Living with HIV/AIDS in Northern Thailand 

Dear Dr. Yanagisawa:

I'm pleased to inform you that your manuscript has been deemed suitable for publication in PLOS ONE. Congratulations! Your manuscript is now with our production department. 

Kind regards, 

on behalf of

Dr. Aftab A. Ansari 

Academic Editor

PLOS ONE